

**Divergent responses of evergreen needle-leaf forests in Europe to the 2020 warm winter**
Mana Gharun[1], Ankit Shekhar[2], Lukas Hörtnagl[2], Luana Krebs[2], Nicola Arriga[3], Mirco
Migliavacca[3], Marilyn Roland[4], Bert Gielen[4], Leonardo Montagnani[5], Enrico Tomelleri[5],
Ladislav Šigut[6], Matthias Peichl[7], Peng Zhao[7], Marius Schmidt[8], Thomas Grünwald[9], Mika
Korkiakoski[10], Annalea Lohila[10], Nina Buchmann[2]
[1] Institute of Landscape Ecology, University of Münster, Germany
[2] Department of Environmental Systems Science, Institute of Agricultural Sciences, ETH
Zurich, Switzerland
[3] European Commission, Joint Research Centre (JRC), Ispra, Italy
[4] Plants and Ecosystems (PLECO), Department of Biology, University of Antwerp, 2610
Wilrijk, Belgium
[5] Free University of Bolzano, Faculty of Agricultural, Environmental and Food Sciences,
39100 Bolzano, Italy
[6] Global Change Research Institute CAS, Bělidla 986/4a, CZ-60300 Brno, Czech Republic,
ORCID: 0000-0003-1951-4100
[7] Department of Forest Ecology and Management, Swedish University of Agricultural
Sciences (SLU), SE-901 83 Umeå, Sweden
[8] Agrosphere (IBG-3), Institute of Bio- and Geosciences, Jülich Research Centre, 52425
Jülich, Germany
[9] Institute of Hydrology and Meteorology, Technical University of Dresden, Dresden,
Germany
[10] Finnish Meteorological Institute, Climate System Research, Helsinki Finland
Corresponding author: Mana Gharun (mana.gharun@uni-muenster.de)

**Abstract**
Relative to drought and heat waves, the effect of winter warming on forest $CO_2$ fluxes during
the dormant season has less been investigated, despite its relevance for net $CO_2$ uptake in colder
regions with higher carbon content in soils. Our objective was to test the effect of the
exceptionally warm winter in 2020 on the winter $CO_2$ budget of cold-adapted evergreen needle-
leaf forests across Europe, and identify the contribution of soil and air temperature to changes
in winter $CO_2$ fluxes in response to warming. Our hypothesis was that warming in winter leads
to higher emissions across colder sites due to increased ecosystem respiration. To test this
hypothesis, we used 98 site-year eddy covariance measurements across 14 evergreen needle-
leaf forests (ENFs) distributed from north to south of Europe (from Sweden to Italy). We used
a data-driven approach to quantify the effect of air and soil temperature on changes in net
ecosystem productivity (NEP) during the warm winter of 2020. Our results showed that the
impact of warming was different across sites, as in the lower altitude and lower latitude sites
positive soil temperature anomalies were larger, while positive air temperature anomalies were
larger in the northern latitude and high-altitude sites. Warming in winter led to a divergent



response across the sites. Out of 14 sites only in 3 sites net ecosystem productivity declined in
winter significantly in response to warming. In addition, we observed that in the colder sites
daytime NEP (that is dominated by photosynthesis) declined with warming of the air in winter,
whereas in the warmer sites daytime NEP increased with warming of the soil. While warming
increases ecosystem respiration, it might not trigger productivity in winter if the soil within the
rooting zone remains frozen. Forests within the same plant functional type category can exhibit
differing reactions to winter warming and to predict their responses accurately it is crucial to
account for variations in local climate, physiology, and structure simultaneously.
**Keywords:** eddy covariance, respiration, productivity, long-term, extremes, carbon flux
**Introduction**
One of the largest sources of uncertainties in understanding how forests can mitigate climate
change is the variation of forest $CO_2$ fluxes in response to extreme climatic conditions. Forests
absorb a large part of anthropogenic $CO_2$ emissions (Friedlingstein et al. 2023), but extreme
climatic conditions compromise the capacity of forests for carbon sequestration (Shekhar et al.
2023). While a large body of research focuses on extreme events during the growing season,
effects of warming winters remain understudied (Kreyling et al. 2019). In northern latitudes
and higher altitudes where evergreen conifers dominate, warming events are especially
pronounced during the winter months (IPCC 2014). In 2020, Europe experienced its warmest
winter on record since 1981 and the largest difference relative to the reference period (1981–
2020) was observed in winter over northeastern Europe (Copernicus Climate Change Service
2020). However, it is not clear yet how such winter warming affected winter $CO_2$ fluxes
particularly where forests are covered by snow and with high soil C content. Understanding the
impact of winter warming on forest net $CO_2$ uptake requires high temporal resolution
observations (sub-seasonal, daily) across many regions, as mechanisms that control forest
carbon fluxes are complex and show different responses to changes in climatic conditions,
depending on the region and forest type.
At the tree level, winter warming could increase $CO_2$ uptake in temperature-limited forests.
While little of this uptake is expected to be allocated to stem growth (Krejza et al. 2022), this
increased activity can impact physiological development of plants that are adapted to long cold
periods. Plant $CO_2$ uptake is controlled by a range of physiological responses to light,
temperature and $CO_2$ concentrations. In addition to these external drivers, physiological factors
(e.g., photosynthetic parameters such as light-use efficiency, maximum rate of electron



transport, maximum carboxylation rate, formation of carbohydrate reserves) and structural
characteristics (e.g., leaf area index) which vary across different evergreen needle-leaf forests
(ENF), directly affect how productivity and $CO_2$ uptake might be affected by warming in winter
(Martinez Vilalta et al. 2016; Stocker et al. 2018).
*Importance of winter period for evergreen needle-leaf forests (ENF)*
Forests adapted to cold environments require a persistent number of days with low temperatures
for building hardiness. Sudden warming during winter months can promote vegetation activity
in response to a condition similar to a "false spring" which can interrupt the cold hardiness
process (Laube et al. 2014). Additionally, increased respiration due to warming can deplete
stored non-structural carbohydrates (NSC) and tree hydraulic functioning (if combined with
drought) and affect tree functioning in spring (Sperling et al. 2015). Winter warming also affects
phenological development of trees and increases the chance of photo-oxidative frost damage
during earlier stages of the growing season (Gu et al. 2008; Chamberlain et al. 2019). All of
this would compromise the capacity of the forest for $CO_2$ uptake throughout the year (Desai et
al. 2016).
Environmental cues such as temperature, photoperiod, and light quality control a network of
signalling pathways that coordinate cold acclimation and cold hardiness in trees that ensure
survival during long periods of low temperature and freezing (Öquist and Hüner 2003;
Ensminger et al. 2006). These signalling pathways include the gating of cold responses by the
circadian clock, the interaction of light quality and photoperiod, and the involvement of
phytohormones in low temperature acclimation (Chang et al. 2021). Soluble carbohydrates,
including sucrose (most abundant) accumulate in response to low temperatures, starting from
late autumn throughout winter (Strimbeck & Schaberg 2009; Chang et al. 2015). Persistent
uninterrupted cold periods thus play an important role in forming the photosynthetic capacity
of the trees and their functioning under extreme climatic conditions. Experimental evidence
from temperature-sensitive conifers shows that warm spells in winter can induce premature
dehardening of buds, and result in stunted shoot development in the following spring (Nørgaard
Nielsen & Rasmussen, 2008). In addition to damage from frost, earlier dehardening can
potentially affect the capacity of trees to cope with a range of extreme climatic conditions such
as cold spells, drought and heat waves.
*Effect of warming on forest carbon fluxes*
Forest net ecosystem productivity (NEP) depends on the balance between gross ecosystem $CO_2$
uptake (gross primary productivity, GPP) and ecosystem respiration (Reco). Both these flux
components are highly sensitive to climate drivers (e.g., air and soil temperature, solar



radiation), and thus when canopy structural changes from one year to another are negligible,
the interannual variations can be predominantly explained by changes in the climatic conditions
(Hui et al. 2003). Net ecosystem productivity can increase or decrease with changes in air
temperature. In temperature-limited ecosystems for example, increase in air temperature
increases photosynthesis which leads to a larger gross productivity and potentially increased
net $CO_2$ uptake (if respiration does not increase more). However with warming and increased
temperatures, respiration (autotrophic and heterotrophic) can also increase, and the balance of
this with changes in gross productivity could lead to an increase, no change, or a reduction in
net $CO_2$ uptake (Gharun et al. 2020). In the presence of winter warming, despite more
favourable conditions for photosynthesis, factors such as water stress or photoinhibition caused
by high photon flux densities in combination with low air temperatures could downregulate
photochemical efficiency and negatively affect net photosynthesis which could decline gross
primary productivity (Troeng and Linder 1982).
The temperature sensitivity of ecosystem respiration regulates how the terrestrial $CO_2$
emissions respond to a warming climate. Within naturally occurring temperature ranges,
ecosystem respiration (sum of autotrophic and heterotrophic respirations) typically shows an
exponential increase with  temperature (Lloyd and Taylor 1994). While previous studies have
shown an increase in $Q_{10}$ (times of increased soil respiration with a 10 °C increase of
temperature) with decrease in site mean temperature (e.g.,  Chen et al. 2020), the temperature
sensitivity of ecosystem respiration incorporates both the direct response of ecosystem
respiration to temperature and indirect influences from other climatic and physiological
variables such as moisture, leaf area index, photosynthate input, litter quality, microbial
community (Reichstein et al. 2002; Fierer et al. 2005; Lindroth et al. 2008; Migliavacca et al.
2011; Karhu et al. 2014; Collalti et al. 2020). These factors change across species composition
and climatic regions and make predicting changes in forest carbon fluxes in response to
warming challenging.
The winter of 2019-2020 was reported as the hottest on record (1981-2022) across Europe
(Copernicus Climate Change Service/ECMWF). When compared to the average conditions, up
to 45 less winter ice days were detected in eastern Europe Russe (C3S/KNMI). In Finland, for
example, the average air temperature for January and February was over 6 degrees higher than
the 1981-2010 mean (Copernicus Climate Change Service/ECMWF). In this study we
investigated how the exceptionally warm winter of 2019-2020 affected ENFs in Europe. Our
objectives were to:



1) evaluate the relative change in air and soil temperature during the winter 2019-2020,
compared to a 6-year reference period of 2014-2019, 2) quantify the relative changes in the
winter $CO_2$ fluxes across coniferous sites with available ecosystem-level $CO_2$ flux
measurements, and 3) identify the contribution of climatic drivers (air temperature, soil
temperature, solar radiation) to changes in $CO_2$ fluxes during the warm winter. Our hypothesis
was that warming in winter leads to a larger negative effect on net $CO_2$ balance (i.e., higher
emissions) across colder forests. We addressed these objectives and tested our hypothesis by
exploring ecosystem-level $CO_2$ fluxes measured with eddy covariance over 98 site-years in 14
evergreen needle-leaf forests distributed from the Boreal to the Mediterranean regions.

**Material and Methods**
*Site description*
We selected 14 evergreen needle-leaf forests where continuous $CO_2$ fluxes and meteorological
measurements were available for at least six years until the end of 2020. Selected sites were
located from the northern to the southern edge of ENF forest distribution in Europe (Figure 1).

**Figure 1** Location of the 14 Evergreen Needleleaf Forest (ENF) sites included in this study.
Base-map is the MODIS Land Cover Product (MOD12Q1, 500m spatial resolution) showing
the distribution of ENFs in Europe in 2020. Elevation of the sites ranges from 4 m a.s.l. (IT-
SR2) to 1735 m a.s.l. (IT-Ren).

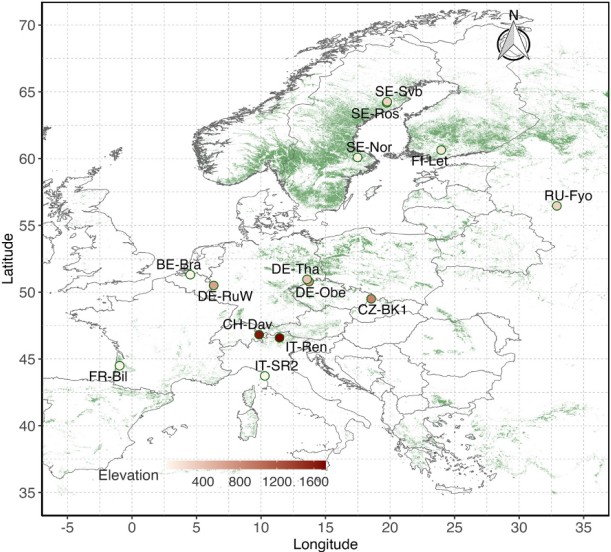




The most northern site studied is located in Sweden at 64.2 °N (SE-Svb) and the most southern
site in Italy at 43.7 °N (IT-SR2). Mean annual air temperature varies between 1.8 °C (in SE-Ros
and SE-Svb) and 15.4 °C (in IT-SR2) across sites. Mean annual total precipitation varies from 527
mm (in SE-Nor) to 1316 mm (in CZ-BK1). Elevation ranges from 4 m a.s.l. (IT-SR2) to 1730 m
a.s.l. (IT-Ren). Table 1 summarizes the description of sites including their dominant canopy
species.

**Table 1** Description of the 14 ENF study sites. Mean annual temperature and total precipitation
refer to the 2014-2019 period. Mean number of days with snow cover for each site is based on
the MODIS satellite observations. Sites are listed in a decreasing order in the mean annual
temperature.

| Site ID | Lat. (º) | Long. (º) | Altitude (m a.s.l.) | Canopy species (dominant first) | Mean annual temperature (ºC) | Mean annual precipitation (mm) | Nr days with snow cover |
|---|---|---|---|---|---|---|---|
| IT-SR2 | 43.702 | 10.290 | 4 | *Pinus pinea* | 15.7 | 950 | 0 |
| FR-Bil | 44.493 | -0.956 | 39 | *Pinus pinaster* | 14.1 | 930 | 11 |
| BE-Bra | 51.307 | 4.519 | 16 | *Pinus sylvestris* | 11.5 | 750 | 20 |
| DE-Tha | 50.962 | 13.565 | 385 | *Picea abies* | 10.2 | 843 | 41 |
| DE-RuW | 50.504 | 6.331 | 610 | *Picea abies* | 8.7 | 1250 | 50 |
| DE-Obe | 50.786 | 13.721 | 734 | *Picea abies* | 7.4 | 996 | 90 |
| SE-Nor | 60.086 | 17.479 | 45 | Mixed (*Pinus sylvestris*, *Picea abies*) | 7.2 | 527 | 89 |
| CZ-Bk1 | 49.502 | 18.536 | 875 | *Picea abies* | 7.1 | 1316 | 71 |
| RU-Fyo | 56.461 | 32.922 | 265 | Mixed (*Picea abies*, *Betula pubescens*) | 6.1 | 711 | 58 |
| FI-Let | 60.641 | 23.959 | 111 | Mixed (*Pinus sylvestris*, *Picea abies*, *Betula pubescens*) | 5.9 | 627 | 99 |
| IT-Ren | 46.586 | 11.433 | 1735 | *Picea abies* | 5.5 | 809 | 112 |
| CH-Dav | 46.815 | 9.855 | 1639 | *Picea abies* | 4.8 | 1062 | 139 |
| SE-Ros | 64.172 | 19.738 | 160 | *Pinus sylvestris* | 4.0 | 614 | 102 |
| SE-Svb | 64.256 | 19.774 | 267 | Mixed (*Pinus sylvestris*, *Picea abies*, *Betula pubescens*) | 3.2 | 614 | 106 |


*Dataset*
We used the Warm Winter 2020 eddy covariance dataset processed with FLUXNET pipeline
(compatible with the FLUXNET2015 collection) in this study (Warm Winter 2020 Team, & ICOS
Ecosystem Thematic Centre, 2022); https://www.icos-cp.eu/data-products/2G60-ZHAK)
(Pastorello et al. 2020). We included the analysis of the spring season at each site to account for



the responses immediately after the winter season.  Winter months included December, January,
and February and spring months included March, April, and May. The 6-year reference period was
from 2014 to 2019. This period was selected to have sufficient temporal overlap between the sites.
NEE quality-checked with a constant friction velocity (u*) threshold was used for all sites
(NEE_CUT_REF)(Shekhar et al. 2023). For an easier interpretation, we present net ecosystem
exchange as net ecosystem productivity (NEP = -NEE) where a negative NEP indicates that forest
is a net source, and positive NEP indicates forest is a net sink of $CO_2$ (Chapin et al. 2006).
In terms of climatic variables we selected those that overlapped across all sites during the study
period. These included incoming shortwave radiation ($R_g$), air temperature ($T_{air}$), soil temperature
at 5cm ($T_{soil}$), precipitation and top soil water content. Given that continuous long-term snow depth
measurements were not available at all sites, we used remotely sensed snow depth products to
quantify mean snow depth and snow depth anomalies in winter 2020. The snow depth data were
derived from the simulation of the Famine Early Warning Systems Network (FEWS NET) Land
Data Assimilation System (FLDAS) (McNally et al., 2017). FLDAS data are produced from the
Noah version 3.6.1 Land Surface Model (LSM) at a monthly resolution with a global coverage at
a spatial resolution of 0.1° × 0.1° (approx.10 km × 10 km) (Kumar et al., 2013) and has been used
in the past to study global spatiotemporal patterns of snow depth and cover (Notarnicola 2022).
For snow cover we used MODIS/Terra (MOD10A2) and MODIS/AQUA (MYD10A2) (Hall and
Riggs, 2021) Snow Cover 8-Day L3 Global 500m SIN Grid, Version 6 dataset, which provides
maximum snow cover extent at 8-day temporal resolution and 500m spatial resolution. For each
forest site, we derived average (2014-2019) leaf area index (LAI) from the LAI Collection 300 m
Version 1.1 product (LAI300) provided by the Copernicus Global Land Service (Fuster et al.,
2020). Average LAI was estimated for each site during the mean net $CO_2$ uptake period
(Supplementary Figure 2). Start of the net carbon uptake period was defined as when daily NEP
crosses from negative to positive, and end is the inverse.
*Statistical analysis*
We compared average daily and daytime (when $R_g > 10$ W/m$^2$ and local time 8-18h) means of
each variable (*v*; climate drivers, $CO_2$ fluxes) during the winter and spring of 2020 to the mean
from a 6-year reference period (2014-2019) using a t-test ($p < 0.05$). Daily means of each variable
was calculated only using the measured and good quality gap-filled half-hourly data (variable
quality control = 0 or 1). To understand the major drivers of winter and spring NEP for each forest
site, we derived conditional variable importance ($CVI_v$) of each predictor variable ($R_g$, $T_{air}$, and
$T_{soil}$) based on a random forest regression model (Breiman, 2001). Soil water content was removed





from the drivers analysis because of its negligible effect on the overall model. We tuned the
random forest model by iterating 'ntree' parameter (number of trees to grow) from 100 to 500 with
steps of 50, and 'mtry' parameter (number of variables to try at each split) from 1 to 3 with steps
of 1, and chose the parameter (ntree = 300 and mtry = 2) with the minimum mean square error.
$CVI_v$ accounts for the correlation between the predictor variables, and was calculated using the
*party* R-package (Hothorn et al., 2006). Based on a 7-day moving window (centered on the central
value of the window) we calculated the mean daily (and daytime) NEP, $T_{air}$, $R_g$, and $T_{soil}$. To
compare the $CVI_v$ across sites, for each site we calculated the relative CVI (RCVI) for each
variable as per equation 2.
$RCVI_v\ (\%)\ =\ \frac{CVI_v}{\sum CVI_v} \times 100$           Equation 2
Where $\sum CVI_v$ is the sum of $CVI_v$ of all variables used in the model. We expressed changes in
variable during 2020 ($v_{2020}$) and the reference period ($v_{reference}$) based on its relative anomaly
($\Delta v_r$ ) and absolute anomaly ($\Delta v_a$) as per equations 3 & 4.
$\Delta v_r\ (\%)\ =\ \frac{v_{2020} - v_{reference}}{|v_{reference}|} \times 100$         Equation 3
$\Delta v_a\ =\ v_{2020} - v_{reference}$         Equation 4
To further understand the how (absolute) anomalies of different variables (daytime $R_g$, $T_{air}$, $T_{soil}$)
explained the variation in daytime ΔNEP, we used the RCVI (as per equation 2) derived from
(also) a random forest regression model with hyperparameters *ntree* = 100 and *mtry* = 3 (tuned
for lowest mean squared error), for each site (number of data points at least 80 days). The %
variance explained of the model was based on the out-of-bag estimates.

**Results**
*Warm winter 2019-2020 conditions across different sites*
According to the *in-situ* data, compared to the reference period (2014-2019), winter 2020 was the
warmest winter across 10 sites. In seven sites, the winter also had lower precipitation than normal
(Figure 2, and Supplementary Figure 1). Positive air temperature anomalies in winter 2020 were
larger in the high latitude or high-altitude sites compared to the mid-latitude and low-elevation
sites (Figure 3) with largest anomaly of 4.79 °C in RU-Fyo and lowest positive anomaly of 0.87
°C observed in IT-SR2 (Figure 3). The average number of snow cover days per year was highly



variable across the study sites. (Table 1). The southernmost site studied here (IT-SR2) typically
has no snow cover in winter, while the subalpine forest in Switzerland (CH-Dav) has on average
139 days with snow (Table 1). In those sites with consistent snow cover in winter (11 out of 14
sites) snow depth declined at 9 out of 11 sites during the warm winter of 2020 and reduction was
considerable in FI-Let, RU-Fyo, SE-Nor, DE-Obe, DE-Ruw, and DE-Tha (Figure 4). In SE-Svb,
FI-Let and DE-Obe soil temperature at 5 cm was continuously above the freezing level in winter
2020 (Figure 5), unlike the mean conditions at the sites where soil temperature fluctuates around
zero in winter. Changes in winter temperature were more significant in winter than in spring
(Figure 3), which is the reason why we focus on the effect of winter warming on $CO_2$ fluxes only.
*Effect of climate drivers on winter $CO_2$ fluxes*
The annual net productivity of ENFs varied from being a maximum sink (±sd) of 797 (± 320) g C
$m^{-2}$ $yr^{-1}$ (CZ-BK1) to a maximum source of -311 (± 93) g C $m^{-2}$ $yr^{-1}$ (SE-Nor) during the six-year
reference period (2014-2019) (Table 2). Inter-annual variation in NEP was largest in CZ-BK1 (320
gC $m^{-2}$ $y^{-1}$) and lowest in SE-Svb (35 gC $m^{-2}$ $y^{-1}$) (Table 2). The length of the net $CO_2$ uptake
period was on average 178 days but varied between the sites from 105 days (in RU-Fyo) to 315
days (in DE-Ruw) (Table 2, Suppl. Figure 2). Except FR-Bil and DE-RuW all sites were a $CO_2$
source in winter under reference conditions, however in IT-SR2, the forest shifted from a $CO_2$
source into a $CO_2$ sink in winter 2020 (Supplementary Table 1).
During the warm winter 2020, mean daily NEP (i.e., annual winter $CO_2$ sink or source strength)
changed significantly ($p < 0.05$) in 9 out of 14 sites (BE-Bra, CZ-BK1, DE-Obe, FI-Let, IT-Ren,
IT-SR2, SE-Svb, SE-Nor, RU-Fyo, grouped as the "affected" sites) compared to the 2014-2019
reference period, with changes in both positive and negative directions (Figure 6). For example,
in BE-Bra, DE-Obe, IT-Ren, SE-Svb and FI-Let, the forest became a significantly larger source
of $CO_2$ in winter 2020, while in IT-SR2, SE-Nor, CZ-BK1, and RU-Fyo forest shifted towards
being a smaller source for $CO_2$ in winter 2020 (Figure 6, Supplementary Table 1). IT-SR2 showed
the largest increased daily NEP in winter (346%) and BE-Bra showed the largest negative anomaly
in daily NEP (-97%) (Figure 6). During the warm winter ecosystem respiration (approximated by
nighttime NEP) increased significantly across 10 out of 14 sites (Figure 6). Daytime NEP however
(dominated by productivity) increased significantly with warming in only 5 sites, and mainly in
the warmer sites (Figure 6).
The relative importance results of the random forest regression analysis showed that across tested
variables, $R_g$ generally had the largest control on NEP. However, with decrease in site baseline
(i.e., mean) temperature, the effect of $R_g$ declined (Figure 7). For example, in the three coldest





sites (SE-Svb, CH-Dav, IT-Ren) $R_g$ had a relative importance of 52%, 23% and 41% for the
variations in NEP respectively, while in the three warmest sites (IT-SR2, FR-Bil and BE-Bra) $R_g$
had a relative importance of 73%, 81% and 58% for NEP respectively (Figure 7). Radiation
dominated the effect on winter GPP and temperature dominated the effect on winter respiration
fluxes. Particularly in the colder sites the effect of radiation was the least (Figure 7).


**Figure 2** Winter temperature and precipitation anomalies (x_anom = x-x_mean) in 2020 (between
December 2019 and February 2020) at those sites where winter 2020 was the warmest and driest
relative to winters during the reference period 2014-2019. Precipitation anomalies are converted
to relative change (relative to mean) but temperature changes are in the original unit (°C).
Anomalies are classified in four main classes of "wet-warm", "dry-warm", "wet-cold", and "dry-
cold". Winter 2020 is marked in bold. Symbols are marked in blue and label (year) is displayed
only if precipitation change was larger than 10% and at the same time temperature change more
than 0.5 °C. Sites ordered by increasing mean temperature (SE-Svb coldest and IT-SR2 warmest).

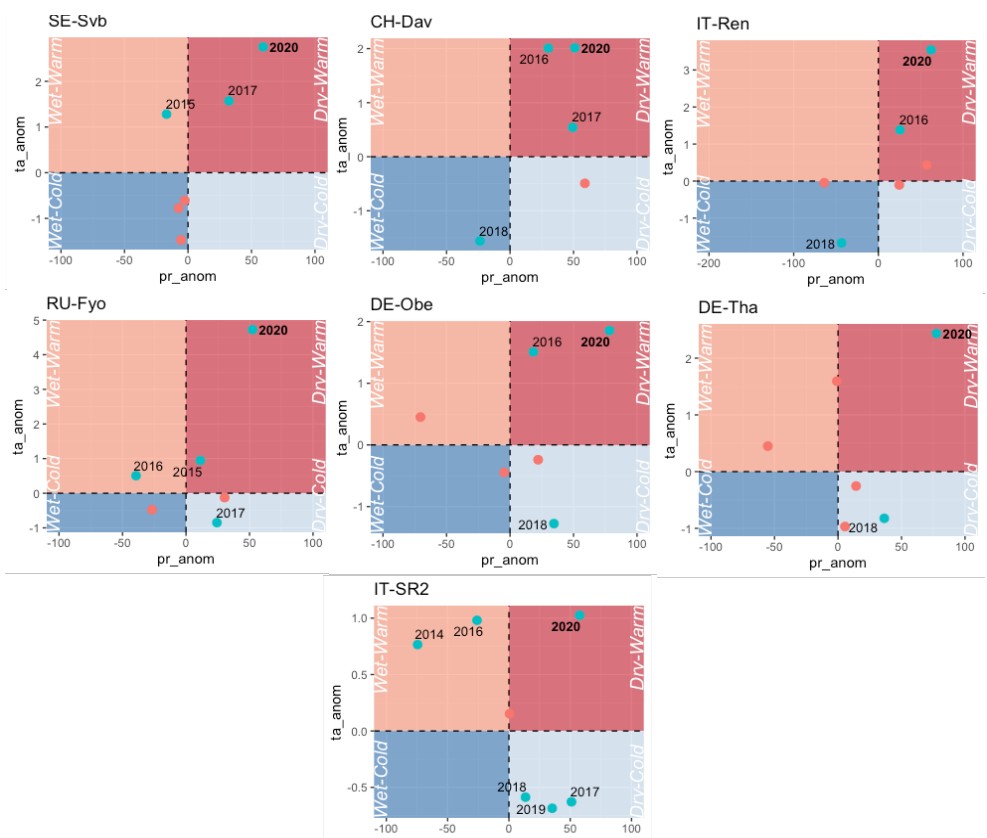




**Figure 3** Seasonal changes in air temperature ($T_{air}$) and soil temperature ($T_s$) in 2020 compared to the 6-year reference period (2014-2019). Asterisk marks where means in 2020 were significantly different from the reference period ($p<0.05$). Anomalies were calculated from daily values. Sites are listed in a decreasing order of mean annual air temperature.

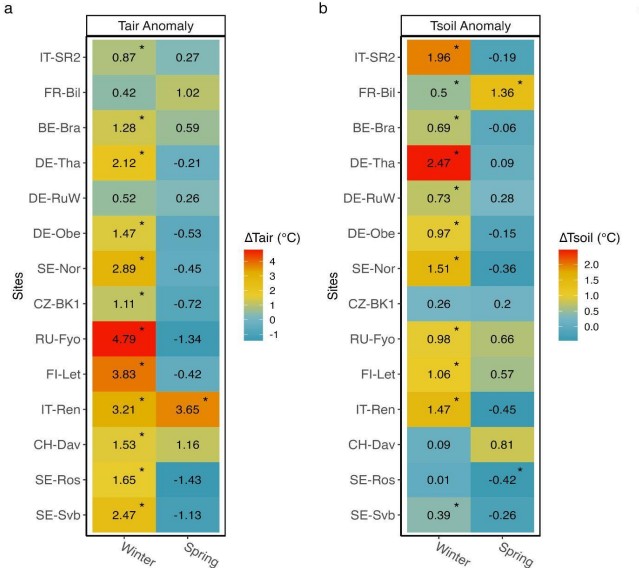

**Figure 4** December to May snow depth changes (cm) in winter 2020 compared to the average winters during the reference period (2014-2019). Note that only 11 out of 14 sites have persistent snow cover in winter. Sites ordered by increasing mean temperature (SE-Svb coldest and DE-Tha warmest).

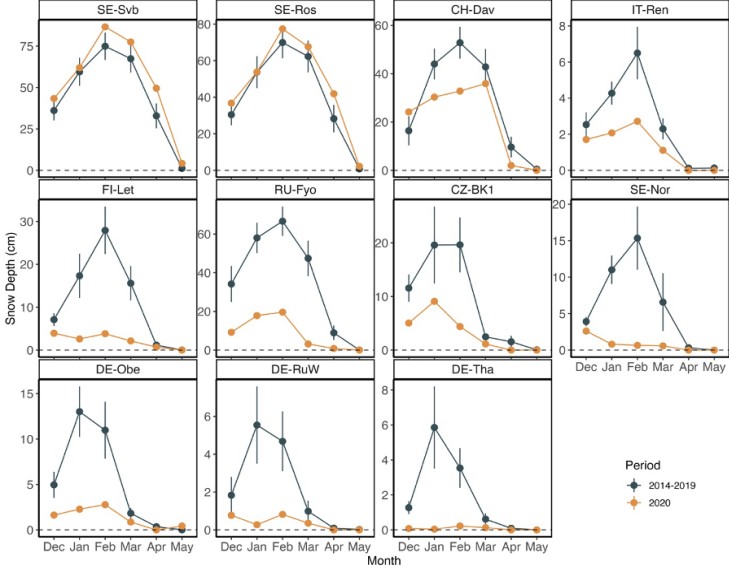





**Table 2** Mean total annual net ecosystem productivity (NEP) and the standard deviation (inter-
annual variation) during the reference period (2014 and 2019). Start of the net carbon uptake
period (SOS, day of year, DOY) is when daily NEP changes from negative to positive and end
(EOS) is the inverse (see Suppl. Figure 2). Sites are listed in a decreasing order in mean annual
air temperature.

| Site ID | NEP (±sd) (g C m$^{-2}$ y$^{-1}$) | SOS (DOY) | EOS (DOY) | Net carbon uptake period (days) |
|---|---|---|---|---|
| IT-SR2 | 197 (±67) | 35 | 200 | 165 |
| FR-Bil | 324 (±103) | 20 | 215 | 195 |
| BE-Bra | 279 (±158) | 95 | 270 | 175 |
| DE-Tha | 484 (±88) | 55 | 305 | 250 |
| DE-Ruw | 597 (±155) | 1 | 365 | 365 |
| DE-Obe | 251 (±147) | 75 | 265 | 190 |
| SE-Nor | -311 (±93) | 90 | 200 | 110 |
| CZ-Bk1 | 797 (±320) | 70 | 310 | 240 |
| RU-Fyo | 25 (±50) | 95 | 200 | 105 |
| FI-Let | -113 (±123) | 100 | 230 | 130 |
| IT-Ren | 675 (±70) | 75 | 305 | 230 |
| CH-Dav | 231 (±139) | 80 | 280 | 200 |
| SE-Ros | 320 (±136) | 95 | 255 | 160 |
| SE-Svb | 163 (±35) | 95 | 240 | 145 |


*Effect of warming on NEP anomalies*
Across the low latitude or low altitude (< 1000 m a.s.l.) sites where NEP changed significantly
in winter 2020 (IT-SR2, BE-Bra, DE-Obe), average NEP anomaly was +75%. In the high-
latitude-high elevation sites where NEP was significantly different in winter 2020 (SE-Nor,
CZ-BK1, RU-Fyo, FI-Let, IT-Ren, SE-Svb) the average NEP anomaly was -8.8% (reduced net
uptake) (Figure 6, Supplementary Figure 5). Average variable explained by the random forest
regression for daytime ΔNEP when abiotic drivers were included in winter was 72% in winter
(Figure 8). Across the affected sites, changes in the air temperature dominated the effect on
NEP anomalies (Figure 8). While FI-Let was affected by a partial cut in 2016 (Korkiakoski et
al. 2019; Korkiakoski et al. 2020), winter fluxes remained relatively stable in all pre- and post-
harvest years as the partial cut affected mostly the summer fluxes (data not shown here).
The relationship between air and soil temperature was stronger than radiation and air
temperature across sites and the relationship between air and soil temperature was stronger in
warmer sites (Table 3). In addition to snow cover, leaf area index and the degree of canopy
closure (directly related to LAI) affect the relationship between air and soil temperature through
a stronger shading of the soil in dense forests. CZ-BK1 had the largest LAI (4.52 ± 0.09 se)



and SE-Ros the smallest (2.59 ± 0.09). FI-Let had the largest inter-annual variation ( ± 0.27)
in LAI and IT-Ren and FR-Bil smallest inter-annual variation (± 0.08) (Table 3).

**Table 3** Pearson correlation coefficient between mean daily incoming shortwave
radiation (Rg), air temperature ($T_{air}$) and soil temperature at 5m ($T_{soil}$) at each site during
the reference period (2014-2019). Sites are ordered by a decreasing mean air
temperature. Leaf area index (LAI) values are shown as mean across the study period ±
standard error of the mean.

| Site ID | $R_g$-$T_{air}$ | $T_{air}$-$T_{soil}$ | LAI ± se |
|---------|-----------------|----------------------|----------|
| IT-SR2 | 0.69 | 0.97 | 3.12 (0.11) |
| FR-Bil | 0.65 | 0.76 | 3.50 (0.08) |
| BE-Bra | 0.67 | 0.92 | 4.42 (0.13) |
| DE-Tha | 0.73 | 0.96 | 4.04 (0.19) |
| DE-RuW | 0.59 | 0.83 | 2.99 (0.22) |
| DE-Obe | 0.72 | 0.94 | 3.69 (0.21) |
| SE-Nor | 0.71 | 0.90 | 3.08 (0.09) |
| CZ-Bk1 | 0.72 | 0.92 | 4.52 (0.09) |
| RU-Fyo | 0.74 | 0.78 | 4.06 (0.14) |
| FI-Let | 0.66 | 0.88 | 3.29 (0.27) |
| IT-Ren | 0.64 | 0.84 | 3.54 (0.08) |
| CH-Dav | 0.63 | 0.87 | 3.25 (0.12) |
| SE-Ros | 0.69 | 0.77 | 2.59 (0.09) |
| SE-Svb | 0.71 | 0.84 | 2.79 (0.12) |

**Figure 5** Soil temperature (at 5cm) changes in winter 2020 compared to the reference period
(2014-2019). Shaded bands around the mean show the 95% confidence interval of mean soil
temperature. Sites are ordered (top and right to left) by increasing baseline temperature (SE-
Svb coldest and IT-SR2 warmest).



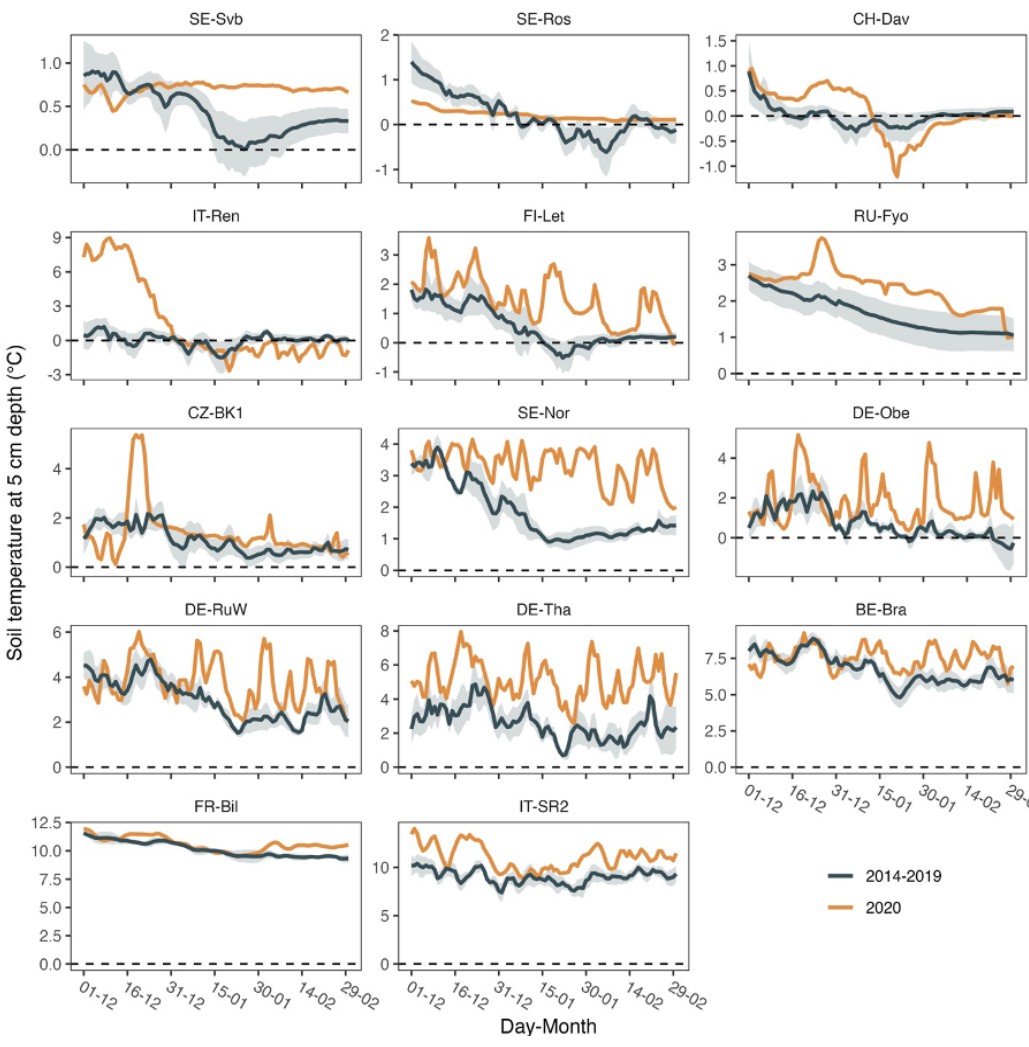




**Figure 6** Relative changes (%) in mean daily, nighttime, and daytime NEP in winter 2020 compared to the 6-year reference winters (2014-2019). Asterisks mark where means in 2020 were significantly different from the reference period ($p <0.05$). Positive NEP change indicates increased net uptake (due to increased uptake or reduced emission) and negative change indicates decreased net uptake (due to reduced uptake or increased emission). Sites are listed in a decreasing mean annual air temperature order.

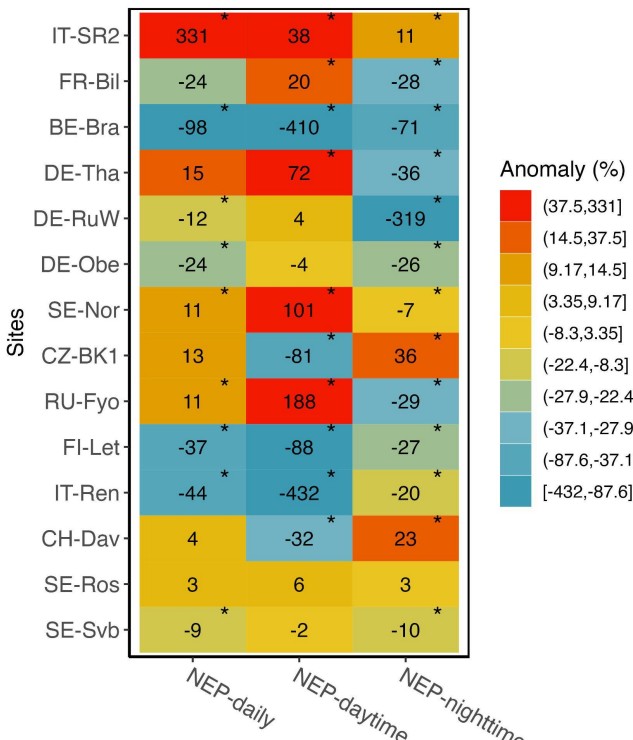



**Figure 7** Relative conditional variable importance (RCVI, %) of three climatic variables for daily winter NEP, GPP and Reco, and the overall variable explained (marked with red triangles) estimated from the random forest regression analysis. The RFR model was trained on winter observations during the reference period (2014-2019). The sites are ordered by decreasing mean annual temperature (top to bottom).

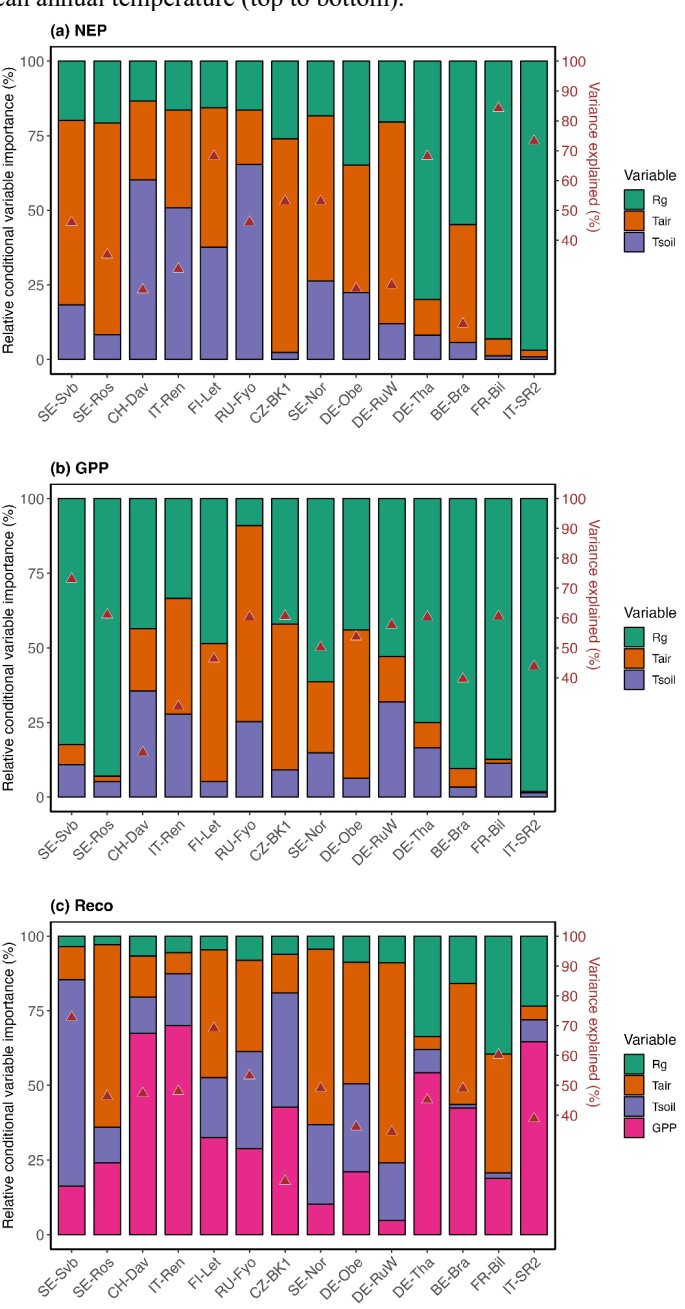





**Figure 8** Comparison of the relative importance of abiotic ($T_{air}$, $R_g$, $T_s$) variables, for NEP
changes (ΔNEP) in winter 2020. $R^2$ of the RFR model that was used to explain the variation
in daytime ΔNEP (i.e., when PPFD > 0) is shown on the secondary (right) y-axis and marked
with red triangles. Sites are ordered by increasing mean air temperature (from left to right).

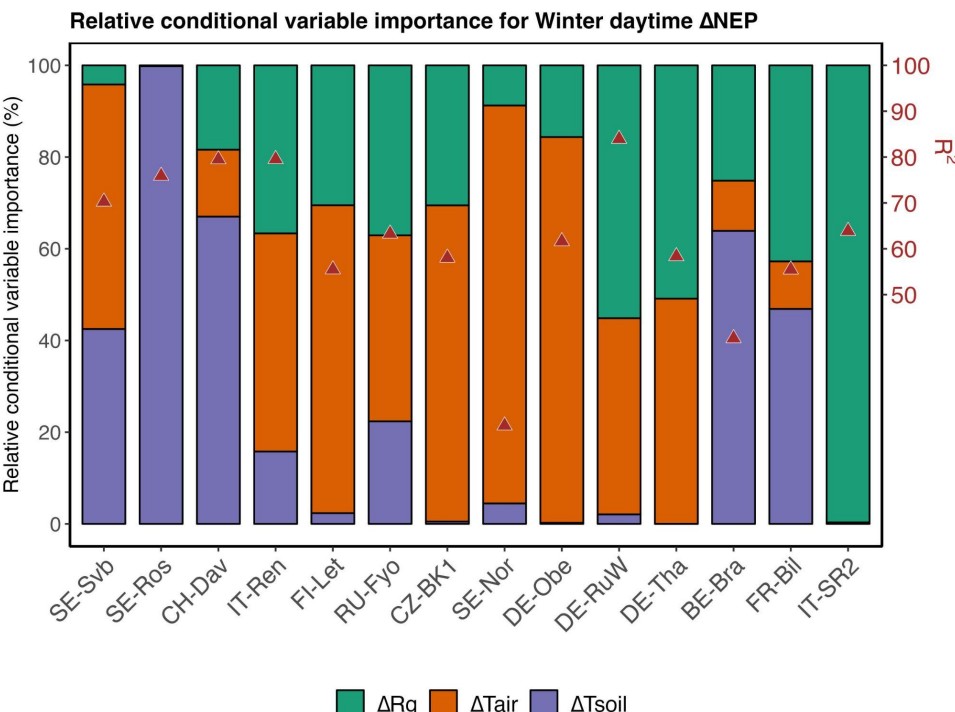

**Discussion**
*Warming of the air and the soil in winter*
We tested how climate variables and $CO_2$ fluxes deviated from a reference period (2014-2019)
during the warm winter of 2020 , across 14 evergreen needle-leaf forest sites distributed from
north to south of Europe (from Sweden to Italy). The sites where winter 2020 was particularly
warm and dry were not clustered in a certain climatic region, however we observed a consistent
pattern that warming of the air was more pronounced in the northern latitude and on high
altitudes sites, while in lower latitudes and altitudes warming of the soil was more pronounced
(Figure 3). While in forests top soil temperature is directly affected by changes in the air
temperature, several underlying processes and properties modify the magnitude of decoupling
of air and soil temperature which could reach up to 10 degrees, depending on the season and
properties of the biome type (Lembrechts et al. 2022). These underlying factors and processes



include for example 1) a vertically complex and horizontally continuous forest structure that leads to higher decoupling of the soil temperature from air temperature, 2) soil moisture content as moisture increases the soil heat storage, 3) insulation by the litter or snow cover, 4) cloud cover, ground surface albedo, and rate of evapotranspiration which collectively affect the radiation balance and energy exchange between the soil and the air, and 5) microtopography that affects the drainage of air (e.g., cool air drains in low-lying areas) (Guan et al., 2009; Lozano-Parra et al., 2018; De Frenne et al., 2021; Gril et al., 2023). Given that in our study the type of forest was similar across sites (all sites were dominated by evergreen needle-leaf forests) and given that our focus was on the warming during the winter season, we attribute the main source of difference in the soil and air temperature to two main factors. First the snow depth that ranged from no snow to over 100 cm across sites (Table 1, Figure 4), and second, differences in forest structure (e.g, LAI) which varied between 2.59 to 4.52 across the sites (Table 3). We observed that the sites with the smaller snow depth showed a larger warming of the soil during the warm winter of 2020 perhaps because the insulating effect of snow cover was weaker here (Friesen et al. 2021) (Figure 2, Figure 4). At the sites where snow depth declined significantly in winter 2020, soil temperatures increased substantially with large fluctuations over the season, whereas in other sites with greater snow depth such soil temperature fluctuations were absent (Figure 5). The link between warming of the air and warming of the soil was also controlled by the canopy structure as we found a significant positive relationship between the two ($p < 0.05$, $r = 0.69$). Although the direct effect of canopy closure on snow distribution, accumulation and melting at different periods was not tested here, it was evident that sites that had a larger LAI also showed a tighter coupling between air temperature and soil temperature as forest canopy closure reduces snow depth (Table 3)(Woods et al. 2006; Gao et al. 2022).

*Winter warming effect on forest $CO_2$ fluxes*

Our general observation was that across sites with a lower mean average temperature (i.e., high altitude or high latitude sites) winter warming was concurrent with increased net $CO_2$ emissions. In the warmer sites however (low altitude or low latitude sites) winter warming also increased the productivity and $CO_2$ uptake (Supplementary Figure 5). This difference can generally be explained by the balance of changes in the warming of the soil versus warming of the air (Bond-Lamberty and Thomson 2010) which affects both soil respiration and tree $CO_2$ uptake. Where soil becomes proportionally warmer and soil temperature reaches above freezing levels, root activity is enhanced and tree productivity responds directly to the increased





air temperatures, and $CO_2$ uptake increases. Warming of the air - if not translated into a direct
warming of the soil– might not enhance productivity if the soil within the rooting zone remains
frozen. In IT-Ren for example where daytime NEP declined significantly in the warm winter,
air temperature increased to over 3.5 degrees more than normal, however soil temperature
remained at freezing levels (Figure 5).
$CO_2$ fluxes are sensitive to changes in both temperature and light (e.g., incoming radiation) and
site baseline climate conditions showed to be a good proxy of how changes in light and air
temperature lead to changes in NEP (Figure 7). There is however evidence that temperature
responses of biochemical processes are a function of plant growth temperature, and not just
instantaneous temperature (Fürstenau Togashi et al. 2018). In addition, response of NEP to
similar temperature can be different across seasons (i.e., an evident hysteresis), depending on
other environmental factors such as solar radiation and soil water content (Niu et al. 2011).
While across different sites sensitivity of NEP to temperature increases with a decrease in site
mean temperature, as site mean temperature increases (temperature is no longer limiting)
radiation becomes a larger constraint on NEP (Running et al. 2004).
Chamber-based observations from boreal forests show that snow-depth and soil moisture affect
temperature sensitivity of soil $CO_2$ fluxes as the freeze-thaw cycles abruptly change the
moisture content of the soil (Du et al., 2013). In that sense, warmer winters can trigger larger
respiration (and availability of nutrients to trees) because of higher $Q_{10}$ of thawed than frozen
soils (Wang et al., 2014), however microbial C limitation can reduce expected increase in
respired $CO_2$, if not countered by greater labile C inputs (Sullivan et al., 2020). In addition,
aboveground productivity increases with increase in temperature (Supplementary Figure 3) and
enhances the autotrophic respiration. Warming in winter also affects the microbial community
that control labile and stable organic carbon decomposition in the soil that would offset
respiration response to temperature and lead to a reduction of soil respiration under warming
(Tian et al., 2021). The magnitude of increase in belowground autotrophic respiration in
response to warming and the supply of labile substrate through rhizodeposition and root
exudate also affects net $CO_2$ fluxes under warming (Nyberg et al., 2020). Decrease in the snow
pack and increased soil freezing has short-term immediate impacts on plant $CO_2$ uptake, but
can also leave a long-lasting negative impact on functioning of trees (Repo et al. 2021).
Particularly sites with prolonged cold winter seasons could be rather negatively affected by the
warming in winter, as we observed through reduced daytime NEP which is an indication of
stress from warming during winter. Trees growing in northern latitudes and higher altitudes
could be more negatively affected by warming in winter as optimal temperatures in trees are



regulated by the short-term changes in temperature, whereas in ecosystems where temperature
fluctuations are seasonally larger, optimal temperature for growth has a broader range (Weng
et al. 2010; Liu 2020).

*Winter tree physiology effect on $CO_2$ fluxes*
Responses of coniferous species to soil warming can vary largely depending on the species'
adaptive traits, the overall ecosystem context, and interactions with other environmental factors
such as precipitation, temperature, and nutrient availability (Dawes et al. 2017; Oddi et al.
2022). The sites we studied here, although all were dominated by evergreen needle-leaf species,
consisted of different canopy species and some sites were dominated by a mixture of species
(Table 1). There can be significant differences in photosynthetic parameters across different
species of evergreen conifers that would affect tree and ecosystem response to warming
(Fürstenau Togashi et al. 2018). The different responses of productivity to increased warming
in ENFs can stem from differences in the quantity (and quality) of stored NSC in the roots, and
the rate at which this C storage is mobilized within the tree during the warm winter (Bansal
and Germino 2009). Warmer temperatures and dry conditions in winter lead to stomatal closure
and depletion of carbohydrate reserves for trees that are adapted to ample precipitation and low
VPD conditions in winter, and this effect leads to reduced $CO_2$ uptake of trees during warmer
winters (Earles et al. 2018).
Low temperature is essential for signals that trigger the synthesis of soluble carbohydrates
involved in osmotic and freezing protection against cold extremes (Chang et al. 2021) that
otherwise impair the Calvin cycle by inhibiting the regeneration of ribulose bisphosphate
(RuBP) and decrease the efficiency of Rubisco carboxylation (Ensminger et al. 2012; Crosatti
et al. 2013). Non-structural carbohydrates (sugar and starch) that are accumulated during the
growing season are utilized in winter to ensure survival of trees (Zhu et al. 2012; Tixier et al.
2020) and failure to develop overwintering defences can cause evergreen conifer needles to
remain susceptible for example to photo-oxidative damage during frost events (Chang et al.
515   2016).
Our results provide the first analysis of the effect of winter warming on $CO_2$ fluxes of evergreen
needle-leaf forests in Europe and point to the importance of understanding multiple underlying
mechanisms that govern $CO_2$ fluxes. Data on the responses of photosynthetic traits on a
timescale that is ecologically relevant (days to years) are scarce, but eddy covariance
observations provide an opportunity for constructing long-term time series of canopy level



521 processes to investigate the effect of extreme climatic conditions across all seasons. We

522 encourage studies that combine long-term observations and plant-level experiments to

523 investigate how changes in the functioning in winter might affect trees' response to extremes

524 that occur earlier in the growing season (e.g., spring frost, spring drought) and to understand

525 the consequences of such extremes for ecosystem carbon uptake.

526

**527 Conclusion**

528 Our study investigated the effect of winter warming on $CO_2$ fluxes of evergreen needle-leaf

529 forests across Europe during the warm 2019-2020 winter. We found significant differences in

530 the impact of warming across sites, with northern and higher-altitude locations experiencing

531 more significant warming of the air, while southern and lower-altitude sites saw greater soil

532 warming. Winter warming influenced forest $CO_2$ fluxes, with daytime Net Ecosystem

533 Productivity (NEP) decreasing in colder sites due to lower soil temperature, while warmer sites

534 experienced increased $CO_2$ uptake. However, responses were not similar across all sites, and

535 factors such as forest structure, and local mean climatic conditions played a role in creating

536 microclimates that buffer or enhance the impact of warming on $CO_2$ fluxes. Understanding

537 these variations combined with tree ecophysiological functioning of cold-adapte ecosystems is

538 crucial for predicting how forests will respond to future winter warming.

539

**540 Acknowledgements**

541 MG acknowledges funding from the Swiss National Science Foundation project ICOS-CH
542 Phase 3 (20F120_198227). TG acknowledges funding from Free State of Saxony (project
543 'Sicherstellung des Treibhausgasmonitorings an sächsischen ICOS-Standorten') and BMBF
544 (project ICOS-D building phase). BG and RM acknowledge the Research Foundation Flanders
545 (FWO) for the support of ICOS research infrastructure. NB acknowledges funding from  the
546 SNF for ICOS-CH Phase 2 (20FI20_173691), and EcoDrive (IZCOZ0_198094). LŠ was
547 supported by the Ministry of Education, Youth and Sports of CR within the CzeCOS program,
548 grant number LM2023048. We acknowledge the ICOS research infrastructure for data
549 provision.


551 *Data availability:* The dataset used in this study is openly available from the ICOS

552 Carbon Portal. https://doi.org/10.18160/2G60-ZHAK


554 *Author contributions:* MG designed the study; MG and AS performed the data analysis;

555 MG wrote the manuscript, and all authors commented on the analysis and contributed

556 substantially to the writing of the manuscript.


558 *Competing interests:* The authors declare that they have no conflict of interests.



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
