# Peer review of "Mana Gharun1, Ankit Shekhar2, Lukas Hörtnagl2, Luana Krebs2, Nicola Arriga3, Mirco Migliavacca3, Marilyn Roland4, Bert Gielen4, Leonardo Montagnani5, Enrico Tomelleri<"

_EGUsphere, 2023_

## Author Comment (AC1)

**Discussion of "Divergent response of evergreen needle-leaf forests in Europe to the 2020 warm winter"**

Gharun et al.

Reviewers' comments are in italic. The Author's responses are marked in blue.

**Author Response to Referee 1**

**1.    General comments**

*The theme holds significant importance, as they aim to assess the impact of a warm winter on CO2 fluxes through the examination of flux tower data. The findings indicate that winter warming triggers a divergent ecosystem response. Additionally, observations reveal that in colder locations, daytime Net Ecosystem Productivity (NEP), primarily driven by photosynthesis, decreases with warming air in winter. Conversely, in warmer sites, daytime NEP increases with soil warming. However, the paper exhibits several big issues:*

*The introductions lack clarity and a well-organized structure. Introductions should avoid incorporating captions.*

Thank you for your feedback. We appreciate your insight regarding the clarity and organization of the introduction. We will take your comments into careful consideration and make necessary revisions to improve the overall structure and coherence of the introduction. Specifically we will improve the clarity of the paper by re-writing the Introduction entirely in this structure:

1) Introducing the main challenge which is understanding the response of forest $CO_2$ fluxes to changes in the temperature and why this understanding is needed,
2) the impact of temperature and radiation on the $CO_2$ flux components (NEP, GPP and RECO),
3) differences in temperature sensitivity of $CO_2$ fluxes in different regions,
4) the extent of reported warming in winter 2020 across different sites
5) the objectives and hypotheses of this study.

*The introduction lacks logical coherence, for example, Lines 78-83 dedicates excessive space to elaborating on physiological responses, which is not the primary focus of this study.*

We will remove text related to ecophysiological responses from the Introduction, and instead add an introduction section about the influence of different environmental factors on the components of NEP (i.e., GPP and ecosystem respiration) and NEP in general, to maintain the focus of the paper.

*Table 3 could be presented graphically.*

After revising our research questions we saw necessary to remove this table, as it no longer fits our objectives.

*The analysis focuses solely on solar radiation and temperature, neglecting the consideration of moisture limitations. RF model should also consider moisture variables.*

Soil water content (SWC) was removed from the drivers analysis 1) because of its negligible effect on the overall model (see more details below), 2) since not all sites had complete measurements throughout the study period, 3) and because generally soil water content measurements should not be trusted at freezing soil temperature levels, and we saw that for several sites soil temperature in winter remained near or below zero (see Figure 5).

The effect of soil water content on the RF model was negligible as we had compared the random forest results once with and once without including SWC. The difference in the variance explained was less than 3% (negligible improvement in results based on the %variance explained of the model).The figure below shows the comparison of the importance of SWC in the random forest model for the 11 sites that had SWC measurements. The dashed line shows the 1:1 comparison of the model with, and the model without SWC.

[Figure]

We will add this information to the Methods section.

*Furthermore, the potential collinearity between soil temperature and air temperature can impact the accuracy and reliability of the model results.*

One of the characteristics of the random forest regression is its robustness against multicollinearity of predicting variables. Decision trees in a random forest model are constructed based on impurity measures known as Gini impurity. When selecting the best split at each node for constructing the decision tree, the algorithm chooses the feature that maximizes the information gain (or minimizes the error), which is a measure of how much the split reduces mse (mean square error) in the target variable. Therefore, even if two correlated variables are available for splitting, the algorithm chooses the one that results in the lowest

error metric (e.g., MSE), which then gives an indication of the importance of the variable (or variable importance) (Breiman 2001). Therefore, random forest is not affected by multicollinearity as much. Furthermore, in this study we do not use the random forest regression model to make predictions, but to understand variable importance by estimating 'conditional variable importance' which calculates variable importance taking into account the collinearity of the predictors (thus 'conditional').

We will add a brief description in the Methods section to clarify this point.

*To enhance the manuscript, a concerted effort should be made to streamline the content, maintain logical progression, and incorporate necessary elements for a thorough analysis.*

Thank you for your feedback.

We will revise the Introduction section thoroughly to improve the logical flow (see our response to an earlier comment please), and will add the missing information in the Methods and Results section (in response to the comments posted by the second reviewer also) and re-write the Discussion section accordingly.

In order to improve the data analysis we will add a section on the analysis of the temperature sensitivity of $CO_2$ fluxes. For this part, we will bin the data by mean air temperature and average $CO_2$ flux for the respective bins. In this way we will compare winter GPP/NEP/Reco with Rg/Tair/Tsoil during a reference winter (2014-2019) and compare that with winter 2020. More details are provided in response to the comments from the Reviewer #2.

---

## Author Comment (AC2)

**Discussion of "Divergent response of evergreen needle-leaf forests in Europe to the 2020 warm winter"**

Gharun et al.

Reviewers' comments are in italic. The Author's responses are marked in blue.

**Author Response to Referee 2**

**1. General comments**

*This is an interesting and ambitious attempt to disentangle the impact of an extremely warm winter on forest productivity (the netto balance between respiration and CO2 uptake). Using high-resolution (temporal) data from multiple forest sites across Europe, linked to locally measured environmental conditions, the authors try to disentangle a) how local microclimatic conditions were different from baseline conditions the years before, and b) how these differences impact winter productivity.*

*While I applaud this effort, the paper might be falling short of answering the actual questions convincingly, perhaps largely due to the extreme complexity of the whole system. Effects are highly site-specific, and the analysis fall short – if I understand all correctly – of showing the direction of trends resulting from the differences in temperature between the warm year and baseline conditions. I'm not sure the answer to the above is more analysis – the paper already has plenty of figures trying to make sense of the complex story – but perhaps a refocus towards figures that synthesize the actual relationship between NEP changes and temperature differences, both within and between sites is needed to bring the story. Also, the figures that are there might need some clarifications to make them more intuitive (see comments at the end).*

We thank you for your feedback. In the revised version we will revise our questions (to make the logic of the paper more clear) and restrict the analysis to answering those questions. In addition we will add figures that synthesize the actual relationship between NEP (and GPP and Reco) changes and temperature differences (see details below).

The questions that we ask are:

1) How much warmer was winter 2020 at each site in terms of air temperature?

2) How did the warming of the air in winter 2020 affect snow depth and soil temperature?

3) How did net carbon uptake change at each site in winter 2020 compared to the reference period?

4) How different was the temperature sensitivity of GPP, Reco, and NEP across different sites and how did this temperature sensitivity of $CO_2$ fluxes change in winter 2020 compared to the reference winter conditions?

Question 1 and Question 2 are answered by Figures 2, 3, 4, 5 that describe the environmental conditions in 2020 compared to the reference period.

Question 3 is answered by Table 2 and Figure 6 that show changes in mean net carbon uptake (NEP)  in winter 2020 compared to the reference winter (2014-2019), in each of the forest sites.

Question 4 is answered by the following figures which we will add to the main body of the manuscript:

NEP (positive values denote a sink) response to air temperature in winter:

[Figure]

NEP response to light in winter:

[Figure]

And the following figure that will be added to the Supplementary Material:

[Figure]

GPP response to light in winter:

[Figure]

[Figure]

In addition we will provide clarification of all figures and make them more intuitive as the reviewer suggested. Please see more specific responses below.

*I have a bunch of more detailed comments below, which I hope identify where for me as a reader the uncertainties arise.*

*L95: why does the risk for photo-oxidative frost damage increase? Is that due to the lost winter hardiness mentioned earlier? Might be good to make the reason explicit.*

The risks of photo-oxidative frost damage increases with winter warming, because warmer winter temperatures can lead to an accumulation of photosynthetically active compounds in plants, and when sudden frost events occur, during periods of high radiation, the combination

of low temperatures and intense sunlight can induce photo-oxidative stress in plant tissues. This occurs because the photosynthetic machinery is still active, but the low temperatures impair the plant's ability to dissipate excess energy, leading to the production (and imbalance) of reactive oxygen species (ROS) that can damage cells and tissues.

Having said this, photochemical damage can also happen in the case of high radiation, low water content in the leaf tissue and low temperature, when photosynthesis and protein turnover become inhibited by low temperatures and when non photochemical, heat dissipation mechanisms are insufficient to deal with excess excitation (hence the negative effect of freezing temperatures after de-hardening) (Anderson & Osmond 1987; Öquist & Huner 2003).

We will add this clarification to the text.

*L99: 'the interaction of light quality and photoperiod': unclear to me what this implies*

Light quality means the type of light in terms of intensity of the red, far red, blue wavelengths, and photoperiod means day length. We meant here that the triggering of cold acclimation in evergreen conifers is not only controlled by temperature but also controlled by daylength, and light properties that change with seasons. The phytochrome system of plants that triggers many different processes is sensitive to the type of light (e.g., to the ratio of red to far-red wavelength).

We will add a clarification to the text.

*L103: 'thus': this word implies that the previous sentences explain why cold periods play an important role in forming the photosynthetic capacity, but to me there is a step missing: why are these pathways resulting in improved photosynthetic capacity. The previous paragraph hints to this, perhaps, by mentioning that a lack of these would result in damage (and thus reduced photosynthesis as a result). But all of this feels a bit implicit and disconnected. Just adding a sentence might already help.*

We will revise this section to make it more clear. The new text will read:

"Environmental cues such as temperature, photoperiod, and light quality control a network of signalling pathways that coordinate cold acclimation and cold hardiness in trees that ensure survival during long periods of low temperature and freezing (Öquist and Hüner 2003; Ensminger et al. 2006). These signalling pathways include the gating of cold responses by the circadian clock, the interaction of light quality and photoperiod, and the involvement of phytohormones in low temperature acclimation (Chang et al. 2021). Soluble carbohydrates, including sucrose (most abundant) accumulate in response to low temperatures, starting from late autumn throughout winter (Strimbeck & Schaberg 2009; Chang et al. 2015). Persistent uninterrupted cold periods thus play an important role in forming the photosynthetic capacity of the trees as warmer winter temperatures increases the chance of photo-oxidative frost damage during earlier stages of the growing season (Gu et al. 2008; Chamberlain et al. 2019) which would compromise the capacity of the forest for CO2 uptake throughout the year (Desai et al. 2016)."

*L104-106: now here you continue with examples. Again, I feel the need for a better structuring of this introduction, disconnecting the theoretical cause-and-effect relationship from the examples.*

Our response to the previous comment will address this comment too.

*L112: might there be a need to define respiration, or can we assume this concept to be sufficiently well-known?*

For the readership of this paper we expect the definition of respiration to be clear. However we will add "emission" to make this clear. The new text will read:

 "Forest net ecosystem productivity (NEP) depends on the balance between gross ecosystem $CO_2$ uptake (gross primary productivity, GPP) and $CO_2$ emission (ecosystem respiration, Reco)."

*L131-132: this sentence is a bit too dense in information for me to fully understand*

We will revise this sentence to make it easier to understand. The new text will read:

The temperature sensitivity of ecosystem respiration incorporates both the direct response of ecosystem respiration to temperature, and indirect influences from other climatic and physiological variables such as moisture, leaf area index, photosynthate input, litter quality, microbial community (Reichstein et al. 2002; Fierer et al. 2005; Lindroth et al. 2008; Migliavacca et al. 2011; Karhu et al. 2014; Collalti et al. 2020). Previous studies have shown that the temperature sensitivity of ecosystem respiration increases with a decrease in site mean temperature  (e.g.,  Chen et al. 2020).

*L133-135: I feel like we're missing some information to make the distinction between direct and indirect effects on respiration clear: what happens in the direct pathway, and can you give some examples on how the other factors are affecting respiration indirectly?*

The direct pathway is that with increase in temperature the metabolic activity of plants and microorganisms increases, leading to higher respiration rates. The indirect pathway is that soil moisture affects the microbial activity and decomposition rates, which in turn influence respiration rates. In moist conditions, microbial activity is larger, leading to increased decomposition and respiration rates. Conversely, in dry conditions, microbial activity slows down, reducing the respiration rates.  LAI affects the amount of plant material available for decomposition. Higher LAI means more plant litter, which increases substrate availability for decomposers and leads to higher respiration rates. The amount of organic matter produced through photosynthesis affects the availability of substrates for microbial decomposition. Higher photosynthate input results in increased carbon availability, stimulating microbial activity and respiration rates. The chemical composition of plant litter influences decomposition rates and microbial activity. Litter with high nutrient content decomposes more readily, leading to higher respiration rates. The composition and diversity of the microbial community in the soil influence decomposition rates and respiration. Different microbial species have varying metabolic capabilities, affecting the efficiency of decomposition and subsequent respiration.

*L140: to me the 'on record' doesn't match too well with the cut-off of 1981, as intuitively I think we have older records than that (although not in that source). Perhaps rephrase to 'in the last four decades'?*

We will replace this term with "in the last four decades".

*L150: soil temperature comes a bit out of the blue here – there has been very little discussion on how it can affect respiration in addition to air temperature. Similar for radiation, for which it's even the first time the word is mentioned.*

We will add a paragraph to the introduction on how the different environmental variables (air temperature, soil temperature and radiation) will affect respiration and GPP fluxes.

*L192: 'the forest'?*

We will rewrite the sentence as suggested.

*L194: what do you mean with climate variables that overlapped? Data availability within your dataset? Or overlapping values?*

We mean data availability within our dataset. We will revise this sentence to make it clear.

*L197: for those sites that measured snow depth, could you make a test of the accuracy of the remotely-sensed snow depth measurements, as this is notoriously hard to get right?*

For most sites we do not have long-term snow depth measurements. We compared the measured snow depth against the remotely-sensed snow depth for one site (DE-Tha) where these measurements were available during the study period. This comparison is shown below. Since we do not have this comparison across all sites we do not include this new figure in the paper and instead address this points within the text (in the Methods section).

[Figure]

*L216: 'were'*

We will rewrite the sentence as suggested.

*L235: there is a 'the' that doesn't seem to fit there*

We will rewrite the sentence as suggested.

*L235: what exactly are these anomalies here?*

As already explained in the Methods, these anomalies are changes in a variable during 2020 ($v_{2020}$) compared to the reference period ($v_{\text{reference}}$) based on its relative anomaly ($\Delta v_r$) and absolute anomaly ($\Delta v_a$) as per equations 3 and 4.

*L247: 'lowest positive anomaly of 0.87°C': what about the even smaller anomalies in Fig. 3, in FR-Bil and DE-RuW?*

Those changes were not significant. We only compare the values when the change was significant at $p < 0.05$. We will make this clear in the text by adding "significantly largest anomaly" and "significantly smallest anomaly".

*L254: first time soil temperature depth is mentioned – feels like more something for the methods*

We will make sure this is mentioned in the Methods section.

*L263: 315 days? Table says 365*

Thank you for pointing out this typo. The correct number is 315. We will correct this in Table 2.

*L272: 'IT-SR2... shifted towards being a smaller source in winter' Is that true? I thought that we saw in Supplementary Table 2 that the value actually showed it turned into a CO2 sink?*

We will rewrite this sentence to avoid future confusion.

*L274-275: why are the values described here different from those in Fig. 6? Are they describing something different? If so, why refer to Fig. 6?*

Thank you for this comment. The correct values are 331% (and not 346%) and -98% (and not -97%) as shown in Figure 6. We will correct the typo in the text.

*L276: 'increased significantly' -> remind the reader that increasing here means a decrease in value? As the previous comments also reflect, this part was pretty hard to follow.*

We will clarify in the text by adding "indicated by a negative anomaly in nighttime NEP" to make sure it is not confusing to the readers.

*L285: 'winter respiration fluxes': remind the reader which panel in the figure that is*

This is shown in Figure 7 panel c that displays Reco as the panel title shows. We will add this info in the text.

*L324: where do we see this result?*

In Figure 6 as the next sentence says. We will add this info here too.

*L325: 'high latitude-high elevation': this is either/or, right, not both together? Unclear from the way it's written*

We mean either/or not both together. We will add this info.

*L329: 'dominated': is the difference that strong? Can you quantify that?*

When sites are ranked by mean temperature, in the 5 colder sites (SE-Nor, RU-Fyo, FI-Let, IT-Ren, SE-Svb) out of 8 affected sites (IT-SR2, BE-Bra, DE-RuW, SE-Nor, RU-Fyo. FI-Let, IT-Ren, SE-Svb) temperature dominated.

*L333: what are the implications of those relationships in Table 3? They feel a bit disconnected from the story now (although, as I ask down below regarding Fig. 7 and 8, they might be critical)*

We will remove this Table as we have revised our research questions and this information no longer helps with our story.

*L336-339: the role of LAI feels a bit like an afterthought now, and the way it's currently analyzed (basically qualitatively rather than quantitatively) makes it hard to build strong conclusions on it. You could at least model the relationship between Tair-Tsoil and LAI?*

This was based on Table 3 which we will remove in the revised version (see previous comment please).

*L435: weaker snow buffering effect often results in lower winter temperatures, as snow tends to buffer against freezing temperatures. This makes sense, as snow is usually present when air temperature is below zero, and then soil temperature stays at zero. You see this in Lembrechts et al – Global maps of soil temperature – that soil temperatures in cold regions are usually higher than air temperatures. Here, you have a reduced snow cover, so the only way in which soil temperature can be higher, is if air temperature is also unusually high (i.e., positive).*

We will remove this statement as the previous sentence already sufficiently points to the link between snow cover and changes in the soil temperature.

*L436-438: can you specify which sites this are? I'm getting a bit lost in trying to link the different figures on snow and temperature.*

At these sites both soil temperature increased and snow depth declined significantly: SE-Svb, IT-Ren, FI-Let, RU-Fyo, SE-Nor, DE-Obe, DE-RuW, DE-Tha. We will mark these sites on Figure 4.

*L449: Supplementary Figure 5 shows very little trend to me (the blue and red dots are basically randomly distributed?*

We will remove this Figure as we have revised our research questions and this information no longer helps with our story.

*L445 and following: in these paragraphs lies one of my main questions. Here indeed you are connecting everything together (soil temperature, air temperature, NEP, …). You have an analysis (the machine learning model) that does this as well, but I am missing figures and/or analyses of the relationships between these things. Now, it requires juggling of all figures and tables to connect everything together, but the machine learning model is only used to show the SIZE of the effect, not the actual relationships themselves.*

*For example, if you say: 'warmer sites however (low altitude or low latitude sites) winter warming also increased the productivity and CO2 uptake' then this should be supported with a figure showing delta productivity/CO2 uptake as a function of delta T in winter and background T, and their interactions (for example, a separate line for the relationship for warmer versus colder sites).*

*Similarly, if you say (L453-454) that when soil temperature reaches above freezing level, CO2 uptake increases, this should be shown by a figure showing daytime NEP as a function of soil temperature.*

*These are just two examples, but this comment is valid throughout the chapter. The main conclusions (the relationships between local climatic conditions and NEP are not really shown in the results, if I'm not mistaken.*

Thank you for the suggestion. Yes, it is true that we show the size of the effects and not the actual relationships. This is because our analysis is based on non-linear decision-tree based machine learning models where depicting the actual direction of the relationship is not possible (as it is in linear models with coefficients).

In order to clarify this part we added two new analysis: 1) analysis of the temperature sensitivity for respiration based on Q10 since RECO is the dominant flux component in winter (new Figure below, error bars show the 95% confidence interval) 2) functional relationships between NEP, GPP and RECO with light, air temperature and soil temperature as written in response to the first comment (see Figures above please).

[Figure]

*L461: how does Fig. 7 show that baseline climate conditions are a good proxy for this? Do you mean that if you order sites from warm to cold that there is a rough trend emerging in the amount of variance explained? If so, then I'm not super convinced that 1) this is a 'good proxy', and 2) that it says anything on 'how'.*

In response to the previous comment we will add a new analysis which will address this point too. We will show the relationship between each of the local climatic conditions and NEP.

*L472: perhaps add half a sentence on what a higher Q10 means in practice for these soils.*

We will add how this means that in these soils a higher Q10 means that soil respiration increases faster in response to warming.

*L474: where do these labile C inputs have to come from?*

Labile C are organic compounds that are simple in structure and highly reactive which makes them easily decomposable during respiration. They come from plant material such as leaf and root litter, root exudates. Seasonal changes in labile C input is thus positively related to NPP (Pausch and Kuzyakov, 2018, Wu et al., 2011, Yin et al., 2013). During winter the warming can increase respiration but the lower NPP (and thus lower labile C input) could limit the increase of respiration (Sullivan et al., 2020).

We will add this information to the text to make it more clear for the readers.

*L475: Supplementary Figure 3 is not mentioned in the results, so it's very hard to link this to the story. In this figure, the differences between 2020 and reference are also very hard to spot. If the story in L445 and following is indeed true, then it should show up somehow in figures correlating NEP to temperature (or better perhaps, delta NEP to delta temperature)*

In the revised version of the manuscript we will quantify the temperature response of NEP in 2020 and compare that with the temperature response during the reference period, via a SHAP value analysis. We will remove Supplementary Figure 3 and add a new figure that shows NEP to air temperature response in 2020 compared to the reference period.

In addition we have added to the Abstract that: "Except the southernmost site, warming declined mean winter NEP across all sites where we observed a significant change to previous years".

*Fig. 2 and Supplementary Fig. 1: x- and y-axis labels are not entirely intuitive, yet not explained*

We will add in the figure caption that "pr_anom" and "ta_anom" are precipitation and temperature anomalies.

*Fig. 3: anomalies are in °C?*

Yes. Temperature anomalies should be expressed in °C.

*Fig. 3: unclear from this figure which sites are high latitude or high-altitude sites (L 246)*

In this figure sites are listed in the decreasing order of mean annual temperature (as mentioned in the figure caption). We will revise the text and instead write:

"Positive air temperature anomalies in winter 2020 were larger in the colder sites (Figure 3)".

*Fig. 6: could you change the color scheme so it is white at zero?*

We will change the color scheme so that zero is marked with white, positive anomaly in red and negative anomaly in blue. The updated Figure 3 and Figure 6 will look as following:

Figure 3:

[Figure]

Figure 6:

[Figure]

*Fig. 7: 'overall variable explained', shouldn't that be variance?*

We will change this everywhere to "variance explained ($R^2$)

*Fig. 7: 'three climatic variables': why is there a fourth one – not mentioned in the legend – for the bottom panel? Even more confusing: it doesn't seem to be mentioned in the main text on Fig. 7 either?*

We will adjust the caption and mention it in the main text as well.

*Fig. 7 and 8: isn't there a correlation between Tsoil and Tair? If so, how can the model decide which of the two explains the variance (and have this sum up to 100%)? This is different from the way I'm used to variance partitioning,*

Yes, there is a correlation (not significant for all sites) between Tsoil and Tair during winter as shown in the Figure below. The correlation is not perfect (varies from 0.07 to 0.82) with both Tsoil and Tair showing a different day-to-day variation, i.e., Tsoil showing a damped variation compared to Tair (also shown in Figure below). We used random forest regression for modelling NEP/GPP/Reco, which uses decision trees to form relationships between dependent (NEP/GPP/Reco) and the independent variables (here Tair, Tsoi and Rg).

Decision trees are constructed based on impurity measures known as Gini impurity. When selecting the best split at each node for constructing the decision tree, the algorithm chooses the feature that maximizes the information gain (or minimizes the error), which is a measure of how much the split reduces mse (mean square error) in the target variable. Therefore, even if two correlated variables are available for splitting, the algorithm chooses the one that results in the lowest mse, which then gives an indication of the importance of the variable (or variable importance). In this study, we used conditional variable importance (which is shown in Figures 7 & 8) as demonstrated by Strobl et al. (2008) which indicates the variable importance taking into account its correlation of the variable with another variable (thus 'conditional'). This variable importance approach of random forest does not provide information about the proportion of variance explained (i.e., variance partitioning) in the target variable but rather quantifies the relative contribution of features to predictive accuracy.

We will include the above mentioned clarification in the 'Statistical analysis' section.

[Figure]

[Figure]

Supplementary Fig. 1: SE-Ros or FI-Ros?

Thank you for pointing out this typo. We will write SE-Ros.

*Supplementary Fig. 4: legend doesn't seem to explain the figure*

Supplementary Fig. 4 shows the performance of the random forest model using a 2-D kernel density scatter plot. We have now modified the legend and the figure caption to indicate the density of the data as shown below.

[Figure]

Figure S4. Density scatter plot showing the performance of the random forest regression model used to explain the variation of wintertime NEP. The average variance explained (across all sites) by the random forest model was 78% (r2 = 0.78).

*Figure numbers are not always in the right order throughout the manuscript, which muddies the water unnecessarily.*

We found the incident that happened and we will fix these and check thoroughly in the revised version.

**References**

Anderson JM and Osmond CB (1987) Shade-sun responses: Compromises between acclimation and photoinhibition. In: Kyle DJ, Osmond CB and Arntzen CJ (eds) Photoinhibition, pp 1-38. Elsevier, Amsterdam

Breiman, Leo. "Random forests." Machine learning 45.1 (2001): 5-32.

Curiel yuste, J et al., 2004. Annual Q10 of soil respiration reflects plant phenological patterns as well as temperature sensitivity. Glob. Change Biol. 10, 161–169. https://doi.org/10.1111/j.1529-8817.2003.00727.x

Högberg, P et al., 2001. Large-scale forest girdling shows that current photosynthesis drives soil respiration. Nature 411, 789–792. https://doi.org/10.1038/35081058

Körner, C., 2013. Plant–Environment Interactions, in: Bresinsky, A., Körner, C., Kadereit, J.W., Neuhaus, G., Sonnewald, U. (Eds.), Strasburger's Plant Sciences: Including Prokaryotes and Fungi. Springer, Berlin, Heidelberg, pp. 1065–1166.

Liu, Q et al., 2006. Temperature-independent diel variation in soil respiration observed from a temperate deciduous forest. Glob. Change Biol. 12, 2136–2145. https://doi.org/10.1111/j.1365-2486.2006.01245.x

Migliavacca, M et al, 2011. Semiempirical modeling of abiotic and biotic factors controlling ecosystem respiration across eddy covariance sites. Global Change Biology 17, 390–409. https://doi.org/10.1111/j.1365-2486.2010.02243.x

Pausch, J., Kuzyakov, Y. (2018) Carbon input by roots into the soil: quantification of rhizodeposition from root to ecosystem scale. Global Change Biol., 24 (2018), pp. 1-12

Ping, J et al., 2023. Enhanced causal effect of ecosystem photosynthesis on respiration during heatwaves. Science Advances 9, eadi6395. https://doi.org/10.1126/sciadv.adi6395

Oquist G, Huner NP. (2003) Photosynthesis of overwintering evergreen plants. Annu Rev Plant Biol. 54:329-55. doi: 10.1146/annurev.arplant.54.072402.115741. PMID: 14502994.

Strobl, C., Boulesteix, AL., Kneib, T. et al. Conditional variable importance for random forests. BMC Bioinformatics 9, 307 (2008). https://doi.org/10.1186/1471-2105-9-307

Sullivan, P.F., Stokes, M.C., McMillan, C.K. et al. (2020) Labile carbon limits late winter microbial activity near Arctic treeline. Nat Commun 11, 4024. https://doi.org/10.1038/s41467-020-17790-5

Tang, J et al., 2005. Tree photosynthesis modulates soil respiration on a diurnal time scale. Glob. Change Biol. 11, 1298–1304. https://doi.org/10.1111/j.1365-2486.2005.00978.x

Wu, Z., Dijkstra, P., Koch, G.W., Penuelas, J., B.A. Hungate (2011) Responses of terrestrial ecosystems to temperature and precipitation change: a meta-analysis of experimental manipulation. Global Change Biol., 17 (2011), pp. 927-942

---

## Author Response (AR3)

**Discussion of "Impact of Winter Warming on CO$_2$ Fluxes in Evergreen Needle-Leaf Forests"**

(Old title: Divergent response of evergreen needle-leaf forests in Europe to the 2020 warm winter)

Reviewer comments are highlighted in black, while author responses are marked in blue. Line numbers correspond to the revised document with tracked changes.

**Gharun et al.**

**17 December 2024**

We would like to thank the editor and the referee for the additional round of feedback and would like to take this opportunity to address the remaining comments.

**Response to associate editor comments**

**Associate editor comments:**

Dear authors,

The remaining reviewer requests several revisions mainly about the clarity of the results and the tests. For example, they question the significance of the decline in NEP in colder sites, which did relate to my comments in the previous round of revisions. They also mention that they were unclear about the hypothesis and wonder about more simplified display of results (scatterplots), recommending adding them to the supplemental information.

Based on the relevant questions about the fundamental results of the paper and its clarity of presentation, I do think this requires another round of revisions by the authors (to be considered only by editors) given that I think it would lead to strengthening the manuscript. Please carefully reply to each comment and consider improving the clarity of results for each comment.

Sincerely,
Andrew Feldman

Response: Thank you for your feedback and for clarifying the points raised by the Reviewer. We understand that the hypothesis as we had phrased it deserves the clarification that the reviewer is asking for. We have now made sure this aspect is clear and provided additional scatterplots that the reviewer is asking for and made adjustments in the text. We hope that all points raised by the reviewer are now clarified. Please see below our response to each comment.

**Reviewer comments**

I am happy to review this paper again. Unfortunately for all parties involved, my main point of confusion still remains, despite the well-appreciated efforts by the authors to clarify things further. Indeed, one major point still requires further clarification for me: the significance of the decline in NEP in colder sites. This issue is mentioned in several parts of the manuscript, but the specific statistical tests and results underpinning the conclusion remain fuzzy.

For example, in line 464, you state: "In colder regions, NEP showed a significant decline in response to winter warming, reflecting this heightened sensitivity (r = 0.66, p < 0.05)." However, it is unclear which test in the results section this refers to. I searched for "0.66" in the manuscript but could not locate it in this context. Could you explicitly connect this statement to the corresponding analysis in the results section?

Response: here we had reported the result of a correlation test. However, this result is removed now (and text is adjusted) because in the test we had included delta NEP across all sites, while the hypothesis is about sites where NEP declined. Thus, we should have tested the correlation across sites with significant negative delta NEP. Please see more details below where we explain how we have addressed this point.

Additionally, what exactly are you testing here? Are you analyzing whether the average ΔNEP across all cold sites (with a clear definition of what constitutes a "cold site") is significantly lower than zero? Or is the analysis limited to the subset of sites that already show a significant change, as implied by lines 383–384?

Response: this was the test of correlation between delta NEP and site mean temperature.

Are you testing the results in Fig. 9b? If so, why do they visually not match this trend?

Response: because it was not testing the results in Fig. 9b.

If it is a test only on the significant sites, wouldn't that risk cherry-picking — analyzing significant changes only for sites that already exhibit them? Or is it a regression through those six sites as a function of winter warming, showing that delta NEP gets lower as delta T gets higher? A clearer explanation of the testing procedure and rationale still feels needed. Since the hypothesis is about effect of warming, first it is important to test the sites where the warming was indeed significant.

Response: since we are discussing sites where NEP changed significantly (as mentioned in line 340) we had to restrict this test to the sites that exhibited a significant change.

I understand that you have opted to present more complex figures in the manuscript, which is a reasonable choice. However, this makes the absence of simpler scatterplots, such as one showing ΔNEP as a function of temperature and/or winter warming, more noticeable. Especially the latter plot, ideally with separate trend lines for cold versus warm regions, would greatly strengthen your conclusion that NEP declined significantly in colder regions with increasing deltaT and not in warmer regions. (UPDATE: I understand now from your response to my question that you thought that 'the hypothesis 'Our hypothesis was that warming in winter will lead to a larger negative effect on net ecosystem productivity (i.e., higher $CO_2$ emissions) across colder forests due to increased ecosystem respiration.' can be answered through a linear model with the interaction between mean annual temperature (to identify colder forests) and deltaT (to identify warming).' was asking for a linear model of the relationship between temperature and deltaT. This was a miscommunication, I wanted one of the relationship between delta NEP and the interaction between temperature and deltaT).

If you have deliberately excluded such simpler scatterplots to safe space, would you at least share them with the reviewers? Adding them to the supplementary materials — where space is not a constraint — could significantly enhance the clarity of your analysis. Simple plots often help bridge understanding and provide essential context for interpreting more complex figures. If these scatterplots support your conclusions, they would greatly strengthen your argument. Conversely, if they do not support the conclusion, it raises important questions about the validity of the statistical test referenced in line 464.

This remains important, I believe, as it remains unclear on what the conclusion that NEP is significantly declining in cold regions in response to winter warming is based on. First, I still not entirely agree that there is more decline in NEP in cold than in warm regions (even though the mean of the one positive and two negative NEP changes for the significantly changing warm regions is positive). When taking the 6 coldest regions, we see there are 3 significant negative ones, one significant positive ones, and two non-significant positive ones. Very similarly, for the 6 warmest regions, on the other hand, we have again 3 significant negative ones and one significant positive one, and one non-significant positive and negative one (Fig. 4, daily).

Second, is then the analysis of the relationship of delta NEP with delta T in cold regions based on the six coldest regions? Or only those that show significant changes in delta NEP?

Is this than a regression based on four points? This could be enough if the trends are clear, but to me it remains important to show. Third, if NEP should be declining with winter warming, shouldn't that mean a negative slope, while in Fig. 9b the slopes of the coldest sites are all positive? Are we supposed to see all this in Fig. 9b?

It could also simply be the case – as you mention in your response to the previous review – that the across-region patterns are not sufficiently strong to make many conclusions about them. But then this should really be stated more explicitly throughout the manuscript, so the reviewer refrains from trying to make sense of these across-region patterns as I'm doing here.

Response: In the previous version of the manuscript, we had looked at the relationship between delta NEP and mean site T across all sites where NEP changed significantly (as implied in line

340). The reported $r = 0.66$, $p < 0.05$ belonged to this test. However, the hypothesis is about "higher emissions" meaning that only sites where delta NEP is negative should have been tested.

Our hypothesis - as mentioned in line 35- is that warming in winter leads to higher emissions across colder sites. We looked into the relationship between magnitude of warming (significant delta T, 5 sites), and site mean temperature, with delta NEP at 6 sites where NEP declined significantly. This is shown in the following scatterplots as the reviewer requested:

[Figure]

[Figure]

**Relationship between mean site temp and delta NEP**

$r^2 = 0.47$ , $p = 0.13128$

However, the relationships were not strong enough to draw a conclusion. In fact, we see the opposite pattern (but not significant) in the relationship between mean site temperature and delta NEP compared to what we had expected. Hence in the revised version of the manuscript we revised all statements where a conclusion about larger decline in colder sites was drawn. This includes:  The statement in the abstract lines (Line 45-46). The statement in lines 406-407. The statement in the Conclusions lines 477-478.

We have also added to the Discussion section Lines 399-404:

"Our hypothesis was that warming in winter will lead to a larger negative effect on net ecosystem productivity (i.e., higher CO2 emissions) across colder forests. While we observed that 1) across most sites winter emissions increased during the warm winter, 2) and that generally emissions in winter increase in response to increase in soil temperature (observed at 7 sites, Figure 7), however we did not find a link between warming of the air and increased emissions that would confirm this general hypothesis."

While this hypothesis was not confirmed, the main body of our Discussion remains unchanged as we discussed study objectives that were directly supported by our results. These objectives were to: 1) evaluate the relative change in air and soil temperature and incoming radiation during the winter 2019-2020, compared to a 6-year reference period of 2014-2019, 2) quantify the relative changes in the winter CO2 fluxes across coniferous sites with available ecosystem-level CO2 flux measurements, 3) teasing apart the contribution of air temperature versus soil temperature versus

solar radiation to changes in CO2 fluxes during the warm winter, 4) test the sensitivity of CO2 fluxes to each of the climatic drivers, and 5) test if the sensitivity of CO2 fluxes to temperature changed during the warmer winter compared to previous years.
We have reported on the results on each of these objectives and shaped the Discussion accordingly.

Some remaining minor comments:
Regarding the hypothesis stated in lines 35–36: It explicitly focuses on ecosystem respiration (ER), but the results and conclusions in the abstract predominantly discuss net ecosystem productivity (NEP). It would be helpful to make the link between ER and NEP explicit. For instance, it is possible that the decline in NEP is driven by changes in GPP rather than ER remaining constant. Clarifying this connection would align the hypothesis with the presented results and enhance the overall coherence of the manuscript.

Response: Given that the tests mentioned above were not significant, our hypothesis was not confirmed and thus this statement does not hold any longer. We have made changes in the text to avoid confusion (see responses above please).

You state in your review that: 'Additionally, we have included new lines (124-130) that discuss how the impacts of soil temperature and air temperature on CO2 fluxes differ'. However, these lines don't mention soil and air temperature separately, just talking about temperature. UPDATE: you seem to describe what you say on L157-161, so that looks fine!

Response: OK, thank you.

L381-382: I mentioned in my previous review that I found the mean of three sites, one positive, one negative, not super meaningful. The fact that this mean is positive seems to get quite some weight by the way it is written here. Can it not be removed? I think it partially contributes to my remaining confusion described above.

Response: Our responses to the previous comments address this point now.

Is Supplementary Figure S10 mentioned anywhere in the text?

Response: Yes, it was mentioned in line 433.